# QuantAgent: Price-Driven Multi-Agent LLMs for High-Frequency Trading

## Abstract

Recent advances in Large Language Models (LLMs) have shown remarkable capabilities in financial reasoning and market understanding. Multi-agent LLM frameworks such as TradingAgent and FINMEM augment these models to long-horizon investment tasks by leveraging fundamental and sentiment-based inputs for strategic decision-making. However, these approaches are ill-suited for the high-speed, precision-critical demands of *High-Frequency Trading* (HFT). HFT typically requires rapid, risk-aware decisions driven by structured, short-horizon signals, such as technical indicators, chart patterns, and trend features. These signals stand in sharp contrast to the long-horizon, text-driven reasoning that characterizes most existing LLM-based systems in finance. To bridge this gap, we introduce **QuantAgent**, the first multi-agent LLM framework explicitly designed for high-frequency algorithmic trading. The system decomposes trading into four specialized agents, *Indicator*, *Pattern*, *Trend*, and *Risk*, each equipped with domain-specific tools and structured reasoning capabilities to capture distinct aspects of market dynamics over short temporal windows. Extensive experiments across nine financial instruments, including Bitcoin and Nasdaq futures, demonstrate that QuantAgent consistently outperforms baseline methods, achieving higher predictive accuracy at both 1-hour and 4-hour trading intervals across multiple evaluation metrics. Our findings suggest that coupling structured trading signals with LLM-based reasoning provides a viable path for traceable, real-time decision systems in high-frequency financial markets. Our code and web interface are publicly available at Anonymous Github Link here for **full reproducibility**.

## 1 Introduction

In quantitative finance, technical analysis treats historical price action as the most immediate and information-dense reflection of market conditions (Pring, 1991). The central premise is that market dynamics, including fundamentals, macro events, institutional flows, and collective sentiment, are ultimately embedded in price movements (Murphy, 1999). Each bar, defined by its open, high, low, and close (OHLC), provides a compact yet universal representation of short-horizon market behavior. This structure enables systematic detection of recurring setups such as trends, reversals, breakouts, and momentum shifts across asset classes ranging from equities and commodities to digital assets (Moskowitz et al., 2012). Under the efficient market hypothesis (Fama, 1970), prices adjust rapidly to public information, making patterns in OHLC bars a natural substrate for short-term prediction without reliance on lagging textual inputs.

Large Language Models (LLMs) have recently demonstrated impressive capabilities in multi-step reasoning, tool use, and interpretable decision-making (OpenAI et al., 2024). These capabilities are directly relevant to quantitative trading (Yang et al., 2023), which heavily depends on integrating heterogeneous signals, applying systematic trading rules, and controlling execution risks. However, most existing LLM-driven financial frameworks operate primarily on textual inputs, such as news articles, social media streams, or earnings reports (Nguyen et al., 2015; Xiao et al., 2025; Zakir et al., 2025). This reliance introduces two major limitations: (i) textual signals typically lag price discovery and are incorporated into markets only after the fact (Chordia et al., 2013), and (ii) such data is noisy, unstructured, and difficult to validate (Liu et al., 2022). Since short-horizon market dynamics are already encoded in OHLC bars, a more direct approach is to align LLM reasoning

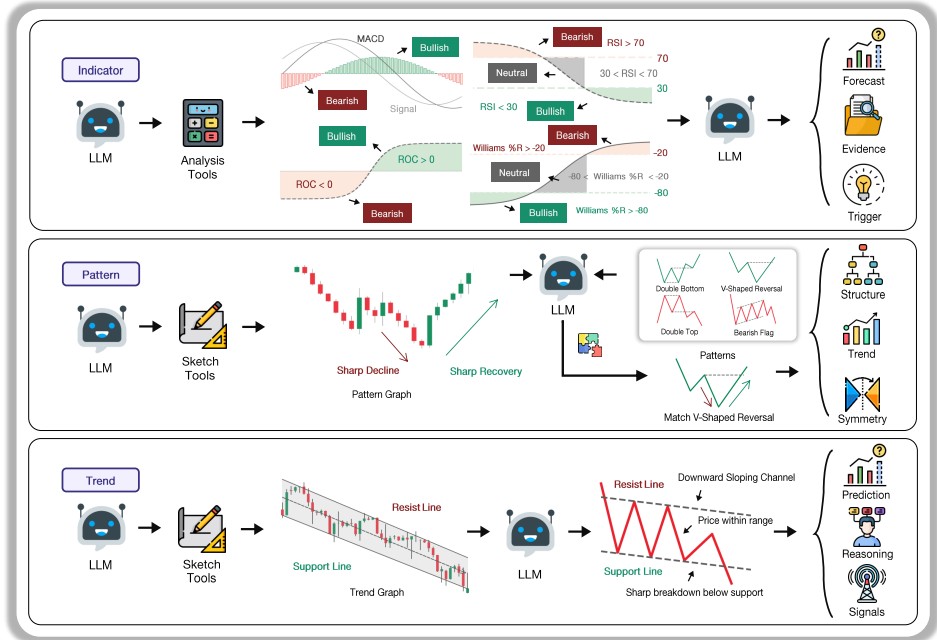

Figure 1: **Workflows of IndicatorAgent, PatternAgent, and TrendAgent.** IndicatorAgent interprets signals from MACD, RSI, ROC, and Williams %R; PatternAgent detects formations such as double bottoms; TrendAgent extracts directional flow via support and resistance channels.

with structured price-based signals. To the best of our knowledge, no prior work has developed an LLM-based framework for high-frequency trading (HFT) that operates directly on OHLC data.

In this paper, we propose **QuantAgent** (Figure 1), the first multi-agent LLM framework tailored to high-frequency algorithmic trading. Specifically, QuantAgent decomposes the trading process into four specialized agents – IndicatorAgent, PatternAgent, TrendAgent, and RiskAgent – each designed to capture a complementary dimension of technical analysis. IndicatorAgent condenses raw OHLC bars into robust technical indicators, providing a noise-resistant summary of recent market behavior. PatternAgent chart formations such as peaks, troughs, and consolidations, leveraging the multimodal reasoning abilities of LLMs (Nison, 2001). TrendAgent identifies directional bias from short-horizon price dynamics, while RiskAgent integrates all signals into a coherent risk–reward profile. Final trade decisions emerge from the interaction of these agents, yielding traceable, language-native rationales that can be inspected alongside execution (Schick et al., 2023).

We evaluate QuantAgent on a multi-asset benchmark spanning commodities, equities, cryptocurrencies, and volatility indices. At 1-hour and 4-hour bar resolutions, QuantAgent consistently outperforms baselines across both directional accuracy and return-based metrics, with particularly pronounced gains in equity markets. Rolling-window validation further demonstrates robust generalization, achieving up to 80% directional accuracy in forecasting short-term price movements. Besides its strong empirical performance, QuantAgent provides natural-language rationales for trading decisions, enabling a degree of traceability and interpretability often missing in traditional algorithmic strategies.

## 2 RELATED WORKS

**Agent-Based LLMs for Financial Decision-Making.** The design of QuantAgent builds on recent work that organizes LLMs into multi-agent systems for financial decision-making. FINCON (Yu et al., 2024) introduces a manager–analyst hierarchy trained via verbal reinforcement learning, while TradingAgents (Xiao et al., 2025) models institutional workflows through agent communication, prioritizing interpretability over the low-latency, price-driven logic required in high-frequency trading (Tumarkin & Whitelaw, 2001). As a line of work, these systems demonstrate the potential of

LLM-based agents in finance, but their heavy reliance on textual inputs leaves them ill-suited for the structured, low-latency signals required in HFT scenarios. More recently, RD-Agent(Q) (Li et al., 2025) takes a significant step forward by shifting to structured, data-centric signals and automating factor–model co-optimization. However, RD-Agent(Q) remains constrained to daily-resolution strategies and slower research-feedback cycles, making it less suitable for real-time decision-making in high-frequency contexts.

**Quantitative Trading Based on Indicators and Patterns.** Prior to LLM-based agents, quantitative trading systems are predominantly built on technical indicators such as trends, volatility, and momentum for intraday decision-making. Early studies show that nonlinear price patterns can exhibit predictive power (Lo et al., 2000), but subsequent work highlight challenges including overfitting and researcher bias (Chen & Chen, 2016). Momentum strategies (Jegadeesh & Titman, 1993; Moskowitz et al., 2012) are widely adopted to capture trend persistence, while heuristics such as Elliott wave theory (Prechter, 2005) and curated pattern libraries attempt to model higher-order market structures. Although these indicator- and pattern-based approaches are interpretable and computationally efficient, they often struggle in volatile or noisy environments, undermining their effectiveness in high-frequency settings. These limitations motivate us to design a framework that fuses structured price signals with LLM-based reasoning, enabling more adaptive and interpretable trading systems.

## 3 QUANTAGENT

To bridge the gap between traditional high-frequency quantitative trading and recent advances in multi-agent LLM systems, we introduce QuantAgent, a collaborative framework for low-latency market decision-making. QuantAgent integrates classical technical analysis with prompt-structured LLM reasoning, enabling modular and interpretable financial intelligence. Built on LangGraph (LangChain, 2025), the system simulates the workflow of institutional trading desks, where specialized agents execute distinct analytical roles to support rapid and coordinated decision-making.

In contrast to prior LLM-based frameworks that incorporate external sources such as news or social media sentiment, QuantAgent operates solely on price-derived market signals. This design choice reflects the efficient market hypothesis, which posits that asset prices incorporate available information by aggregating the actions and beliefs of market participants over time (Murphy, 1999). By grounding analysis exclusively in OHLC data and technical indicators, QuantAgent avoids the latency, noise, and unpredictability of textual sources, while remaining fast, interpretable, and directly aligned with the demands of high-frequency trading.

The system decomposes trading into four specialized agent, IndicatorAgent, PatternAgent, TrendAgent, and RiskAgent, that communicate through structured prompts. Each agent captures a complementary perspective on short-horizon market dynamics: numerical indicators, geometric patterns, directional momentum, and integrated decision-making. In the following subsections, we describe the design of each agent in detail and formalize the technical components underlying our framework.

---

**Algorithm 1:** Slope-aware trend detection over a candlestick sequence $P_{0:T-1}$

**Input:** $P_{0:T-1}$, $N$, $\tau$

1 **for** $t = N - 1$ **to** $T - 1$ **do**
2     Fit OLS on highs/lows to get $m_r, m_s$;
3     $\kappa_t \leftarrow (m_r + m_s)/2$;
4     **if** $\kappa_t > \tau$ **then** Trend $\leftarrow$ Uptrend;
5     **else if** $\kappa_t < -\tau$ **then** Trend $\leftarrow$ Downtrend;
6     **else** Trend $\leftarrow$ Sideways;
7 Render chart $\mathcal{K}_t(P_t, \kappa_t, \text{Trend})$;

---

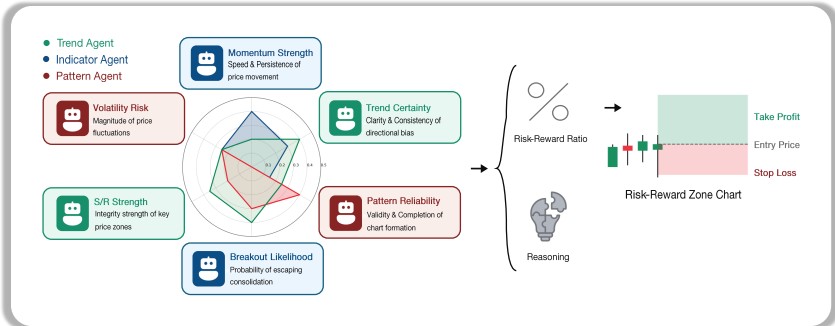

Figure 2: **Workflow of RiskAgent.** Signals from Trend, Indicator, and Pattern are aggregated into a radar chart. DecisionAgent predicts with stop-loss and take-profit.

## 3.1 INDICATORAGENT

**IndicatorAgent** constitutes the initial analytical module of our framework, responsible for transforming raw OHLC sequences into structured quantitative signals (Figure 1). In high-frequency trading (HFT), where decisions must be executed under strict latency constraints, technical indicators provide compact representations that highlight shifts in market momentum and sentiment. Formally, this process can be viewed as a mapping from price tuples to an interpretable signal space, $(O, H, L, C) \mapsto S$, where $S$ denotes a set of derived features summarizing short-horizon market dynamics (Lo et al., 2000). By abstracting low-level price data into high-level features, IndicatorAgent facilitates fast and interpretable downstream reasoning.

To achieve this, it converts raw OHLC values into a compact set of informative technical indicators. Specifically, IndicatorAgent uses five widely used technical indicators to extract signals from market data. RSI captures momentum and flags overbought or oversold zones (Wilder, 1978), while MACD tracks convergence or divergence between short and long term price trends (Appel, 2005). RoC measures the speed of price changes (Murphy, 1999), STOCH identifies turning points based on recent highs and lows (Investopedia, n.d.), and WILLR detects price drops from recent peaks to signal possible reversals (Williams, 2011).

Together, these indicators capture both short-term volatility and longer-term momentum. IndicatorAgent integrates them into a structured summary that reflects current market conditions. Rather than applying fixed rules, it interprets signals in context such as highlighting dynamics like momentum shifts, overbought or oversold zones, and sudden reversals (Murphy, 1999). This contextual reasoning enables DecisionAgent to act on timely, relevant insights, improving responsiveness in fast-moving trading environments (Lo et al., 2000).

### 3.1.1 PATTERNAGENT DESIGN

While IndicatorAgent offers useful signals, these numerical indicators can become unclear—especially when price movement stalls or enters a new regime (Murphy, 1999; Lo et al., 2000). To address these limitations, our PatternAgent introduces a more visual and structural multimodal reasoning perspective (Figure 1).

Upon receiving a request to analyze market patterns, PatternAgent first utilizes LLM-binded tools to generate clear, simplified candlestick charts directly from raw price data. These visualizations present recent market behavior without explicitly highlighting specific geometric shapes or details. Our agent, instead, automatically detects essential visual features from price movements, such as significant highs and lows, structural symmetry, and potential reversal formations,effectively capturing key visual patterns used in technical analysis (Wang et al., 2023).

Using this information, PatternAgent compares the current market structure to an extensive library of well-known patterns described in clear language. Each pattern in this library includes concise yet detailed descriptions of its visual form and the market behaviors it typically signals. Through this comparison, PatternAgent identifies the most plausible match and evaluates its relevance to

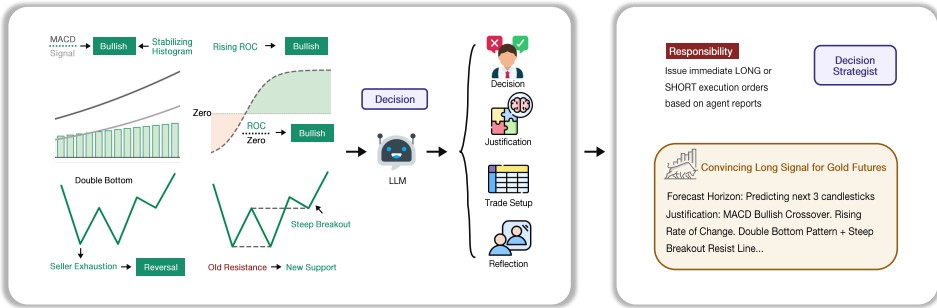

Figure 3: **Workflow of DecisionAgent.** LLM summarizes upstream signals and consolidates them into structured outputs: direction, reasoning, trade setup, and post-trade reflection. It then formulates an executable order with rationale, ready for market submission.

the current context. This process blends visual understanding with language-based reasoning to recognize patterns and assess whether their context—such as preceding trends or volatility—makes them likely to signal a meaningful price move.

By translating complex visual signals into concise and interpretable summaries, PatternAgent plays a key role in bridging raw chart data and high-level reasoning, allowing the system to integrate visual structure into its overall market understanding (Lo et al., 2000).

### 3.1.2 TREND AGENT DESIGN

Canonical chart patterns, such as double bottoms or flags, can be reliably interpreted only when evaluated within the context of a well-defined trend (Pring, 1991). By tracking both the direction and steepness of price movements over time, TrendAgent provides a structural representation of trend dynamics, clarifying whether a detected pattern is consistent with the prevailing trend, signals a potential reversal, or indicates a phase of non-directional price congestion (Elder, 2002).

As shown in Figure 1, TrendAgent generates an annotated K-line chart $\mathcal{K}_t$, which includes a trend channel $\mathcal{C}_t$ that captures the price trajectory through two fitted lines: the upper resistance line $\mathcal{R}_t(x) = m_r x + b_r$, drawn through recent local highs, and the lower support line $\mathcal{S}_t(x) = m_s x + b_s$, drawn through recent local lows (Lo et al., 2000). The trendlines, computed using ordinary least squares regression as outlined in Algorithm 1, serve to characterize price direction, strength, and potential breakout zones. The average slope $\kappa_t = \frac{m_r + m_s}{2}$ provides a basic estimate of directional drift. However, reliable trend classification requires more than just the slope sign, as market noise can obscure short-term movements.

To address that, TrendAgent examines the geometric relationship between the lines—such as parallel upward slopes indicating strong trends, or converging lines forming a wedge that suggests indecision or accumulation. These structural cues allow the agent to reason about not just direction but also the confidence and stability of the trend. They work in conjunction with shape-based patterns and momentum signals identified by other agents, improving decision-making and reinforcing signal coherence across the system (Kirkpatrick & Dahlquist, 2015).

### 3.1.3 RISK AGENT AND DECISION AGENT

RiskAgent, shown in Figure 2, translates technical insights into risk-aware trade boundaries, reflecting the central role of risk control in real-world trading. Instead of signal generation, RiskAgent integrates other agents' output into a unified risk-reward framework. It sets a fixed stop-loss value $\rho = 0.0005$ to account for short-term volatility and computes a take-profit level $\mathcal{R} = r \cdot \rho$, where $r \in [1.2, 1.8]$ is predicted by the LLM. This forms a structured decision zone bounded by stop-loss, entry, and take-profit levels. Within this zone, the agent reasons over signal quality and predefined risk exposure to ensure consistent and informed actions. By aligning domain knowledge with real-time constraints, RiskAgent grounds high-level analysis in practical execution under uncertain and fast-moving market environment.

The final stage of the framework is DecisionAgent, which functions as the reasoning and execution layer. As illustrated in Figure 3, it integrates the outputs of upstream agents to decide between a LONG or SHORT position. Since holding is not permitted, the agent forecasts short-term market direction over the next three candlesticks and generates actionable decisions aligned with the aggregated signals from the overall system (Kissell, 2013).

Specifically, DecisionAgent takes in an aggregated signal state from IndicatorAgent, PatternAgent, and TrendAgent, and outputs a structured trading decision comprising the predicted direction (LONG or SHORT), a concise justification, and a risk–reward ratio conditioned on market context (Kissell, 2013). The agent integrates heterogeneous evidence and evaluates the consistency across upstream signals, proceeding only when the majority align and are reinforced by confirmations such as indicator crossovers, completed breakout formations, or price interactions with major trend boundaries. This layered reasoning filters out noisy or conflicting inputs and yields confident, high-quality decisions. Consequently, the outputs are not only optimized for execution in high-frequency settings but also more robust and interpretable than those produced by traditional rule-based systems.

## 4 Experiments

We evaluate our QuantAgent framework in a fair and comprehensive manner, with trading decisions generated autonomously without prior demonstrations or supervised fine-tuning. Building on the structured reasoning provided by upstream agents, our system uses only recent candlestick data and basic context (e.g., asset type and time interval) to predict short-term market direction. It then generates clear trade suggestions with human-readable explanations, allowing us to evaluate its performance in realistic settings. The experiment is designed to test the framework's effectiveness in realistic, data-limited environments where fast, adaptive decision-making is required.

**Benchmark Construction and Evaluation Protocol.** To support evaluation, we build benchmarks of 4-hour and 1-hour OHLC data across key asset classes such as cryptocurrencies, equity indices, and commodities. For each asset, 5000 historical bars are collected via a public TradingView data extraction tool API. From this, 100 evaluation segments per asset are sampled, each with 100 consecutive candlesticks—the last three withheld to prevent test-time leakage. Details of benchmark are illustrated in Appendix D.

The system processes agents' analysis to generate a structured trade report containing a directional prediction (LONG or SHORT), a brief textual rationale, and an estimated risk–reward ratio. Among these outputs, the directional decision and risk–reward estimate are used for quantitative evaluation.

**Baselines.** We evaluate four categories of baselines: **(i) Random Methods**, which performs random selection of market trend between LONG and SHORT and risk-reward ratio $\in [1.2, 1.8]$. **(ii) Linear Regression**, which serves as a rule-based baseline. It fits a linear model to a 40-bar window of recent closing prices and use the slope of the fitted line to classify future trend. If the slope exceeds 0, the system predicts LONG. Otherwise, it predicts SHORT. **(iii) XGBoost**, which serves as a tree-based supervised learning model. It uses technical indicators extracted via a public API TA-Lib (TA-Lib Development Team, 2025) including RSI, MACD, and SMA. This model is trained on hundreds of sliding-window sample across 50 randomly selected csv files. The trained model is tested on the rest 50 csv files with the same metrics as other methods. A majority-vote rule is applied across predictions to produce a final LONG/SHORT/HOLD decision where HOLD decision is disregarded during final average calculation. **(iv) QuantAgent**, our LLM-based approach, perform short-term prediction using multi-modal multi-agent cooperation.

**Evaluation Metrics.** To evaluate prediction accuracy, we compare the LLM's directional forecast to the next three candlesticks. For a LONG decision, each candle that closes above the last close counts as a correct hit (max 3); for SHORT, each close below the current close counts. Accuracy is defined as $\alpha = \frac{\mathbb{C}}{\mathcal{T}}$, where $\mathbb{C}$ is the number of correct predictions and $\mathcal{T}$ is the total evaluated. Each test yields a score from 0 to 3, and the average is computed over all samples. This aligns with the Mean Directional Accuracy metric used in forecasting (Pesaran & Timmermann, 2004).

In addition, we evaluate trade outcomes based on multiple Rate of Return (RoR) estimators (Fama & MacBeth, 1973) commonly used in HFT, each is to quantify the profitability of a trade by measuring the relative gain or loss between the entry price and exit. All rate-of-return metrics in our framework,

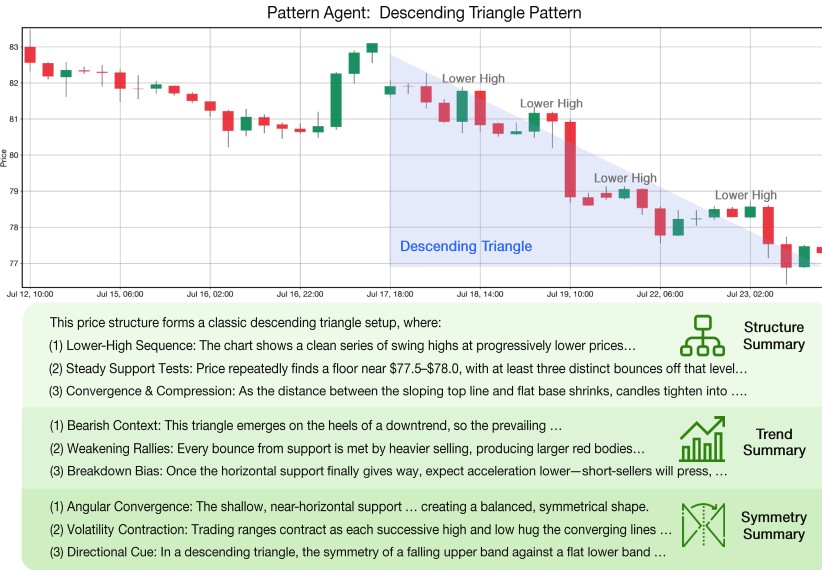

Figure 4: **Case sample of the `PatternAgent` on CL (2024).** The agent extracts swing pivots, fits a declining resistance line through lower highs, and identifies flat support near 78. As the gap narrows, it classifies the formation as a descending triangle and generates three structured summaries: Structure ("lower highs" vs. "flat support"), Trend (bearish breakdown bias), and Symmetry (triangular convergence). Dashed edges and EMA overlays are visual aids only; the classification is derived solely from bar geometry.

including $\mathcal{R}_{cc}$, $\mathcal{R}_{\max}$, and $\mathcal{R}_{\min}$, incorporate risk-constrained execution, simulating realistic stop-loss and take-profit behavior. Specifically, a trade is exited at the first price among the next three candlesticks that hits either the stop-loss or reward threshold. We adopt a fixed stop-loss threshold $\rho = 0.0005$ (i.e., 0.05%), selected to reflect the relatively small fluctuations typical within a short three-candlestick forecast horizon, consistent with prior work (Kissell, 2013). The corresponding reward threshold is determined using the LLM-generated risk–reward ratio $r = \frac{\mathcal{R}}{\rho}$, where $\mathcal{R} = r * \rho$ is the maximum allowed gain and $-r$ is the maximum allowed loss.

$\mathcal{R}_{\max}$ represents the best-case rate of return (RoR) achievable over the next three candlesticks under the current LLM-issued trading decision (either LONG or SHORT). It assumes the optimal exit occurs at the most favorable intra-candle price point, i.e., the maximum high for a long position or the minimum low for a short position. Conversely, $\mathcal{R}_{\min}$ captures the most adverse price movement during the same interval. These two metrics represent a bounded range of maximum profit or loss outcomes under realistic, risk-managed execution (Lo, 2001).

## 5 RESULTS

### 5.1 MAIN RESULTS

In Table 1, we compare our agent-based LLM trader to the three baselines, Random Baseline, Linear Regression(LR) and XGBoost, across eight widely traded markets.

From the table, we draw several key observations: *(i)* Our accuracy outperforms all methods across the eight evaluated markets especially on NQ, where we achieve a 26.5% increase over the random baseline and a clear margin over both LR and XGBoost. *(ii)* Despite the presence of risk caps, our $\mathcal{R}_{cc}$ still achieves the best performance in 7 out of 8 assets, suggesting that our model can consistently capture profitable short-term trends under realistic trading constraints. *(iii)* We obtain the highest $\mathcal{R}_{\max}$ in 6 out of the 8 markets and are nearly tied with the best performer in the remaining two, indicating our system effectively captures potential upside while respecting risk bounds. *(iv)* Similarly, our $\mathcal{R}_{\min}$ shows strong robustness, ranking among the least negative values across most assets. This implies that our method not only captures gains but also limits downside risk effectively.

| Asset | Method | Acc $\alpha$ ↑ | $\Delta\alpha\%$ ↑ | $\mathcal{R}_{cc}$ ↑ | $\mathcal{R}_{max}$ ↑ | $\mathcal{R}_{min}$ ↑ |
|---|---|---|---|---|---|---|
| | Baseline | 45.0 | – | -0.009 | 1.220 | -1.245 |
| | LR | 46.0 | +2.2% | -0.066 | **1.245** | **-1.210** |
| BTC | XGBoost | 45.3 | +0.7% | -0.050 | 1.218 | -1.331 |
| | Our | **50.7** | **+12.7%** | **0.089** | 1.232 | -1.212 |
| | Baseline | 41.0 | – | -0.373 | 0.970 | -1.348 |
| | LR | 54.3 | +32.4% | -0.114 | 1.178 | -1.141 |
| CL | XGBoost | 40.0 | -2.4% | -0.056 | 0.958 | -1.151 |
| | Our | **55.0** | **+34.1%** | **-0.008** | **1.200** | **-1.119** |
| | Baseline | 47.0 | – | 0.048 | 0.755 | -0.793 |
| | LR | 52.0 | +10.6% | 0.149 | 0.790 | -0.725 |
| DJI | XGBoost | 47.3 | +0.6% | -0.020 | 0.874 | -0.660 |
| | Our | **52.3** | **+11.3%** | **0.163** | **0.891** | **-0.649** |
| | Baseline | 51.0 | – | -0.048 | 0.538 | -0.552 |
| | LR | 43.0 | -15.7% | 0.032 | 0.553 | -0.546 |
| ES | XGBoost | 52.0 | +2.0% | -0.182 | 0.440 | -0.644 |
| | Our | **55.0** | **+7.8%** | **0.179** | **0.613** | **-0.485** |
| | Baseline | 46.3 | – | 0.059 | 3.259 | -3.157 |
| | LR | 48.7 | +5.2% | -0.140 | 3.407 | -3.099 |
| VIX | XGBoost | 53.3 | +15.1% | 0.161 | 3.325 | -3.110 |
| | Our | **54.7** | **+18.1%** | **0.458** | **3.872** | **-2.851** |
| | Baseline | 43.7 | – | -0.140 | 0.646 | -0.793 |
| | LR | 48.7 | +11.4% | 0.147 | 0.782 | -0.670 |
| NQ | XGBoost | 47.3 | +8.2% | -0.007 | 0.706 | -0.753 |
| | Our | **55.3** | **+26.5%** | **0.216** | **0.814** | **-0.639** |
| | Baseline | 47.3 | – | -0.048 | 0.930 | -1.017 |
| | LR | 56.0 | +18.4% | 0.175 | 1.113 | **-0.849** |
| QQQ | XGBoost | 52.7 | +11.4% | 0.210 | **1.206** | -0.973 |
| | Our | **59.7** | **+26.2%** | **0.211** | 1.052 | -0.881 |
| | Baseline | 47.3 | – | -0.162 | 0.719 | -0.862 |
| | LR | 59.7 | +26.2% | **0.377** | 0.960 | -0.648 |
| SPX | XGBoost | 60.0 | +26.8% | 0.050 | 0.782 | -0.712 |
| | Our | **63.7** | **+34.6%** | 0.341 | **0.965** | **-0.641** |

Table 1: **Performance comparison across trading symbols.** Results are shown for random(Baseline), Linear Regression(LR), XGBoost, and our QuantAgent. Bold values indicate the best performance for each metric across methods. Upward arrows (↑) denote metrics where higher values are better.

Overall, the results highlight that our approach generalizes well across diverse asset classes in 4-hour time frame, balancing accuracy and return while maintaining robust risk control.

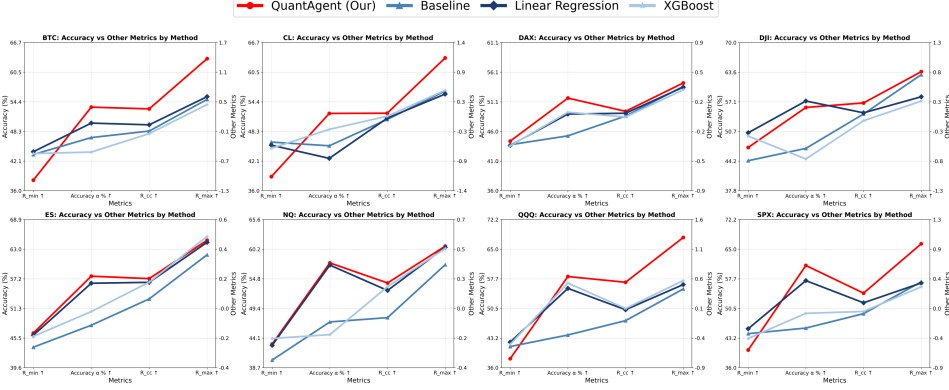

Figure 5: 1-hour performance comparison across eight assets. Results are shown for random (Baseline), Linear Regression, XGBoost, and our QuantAgent. Arrows indicate higher values are better.

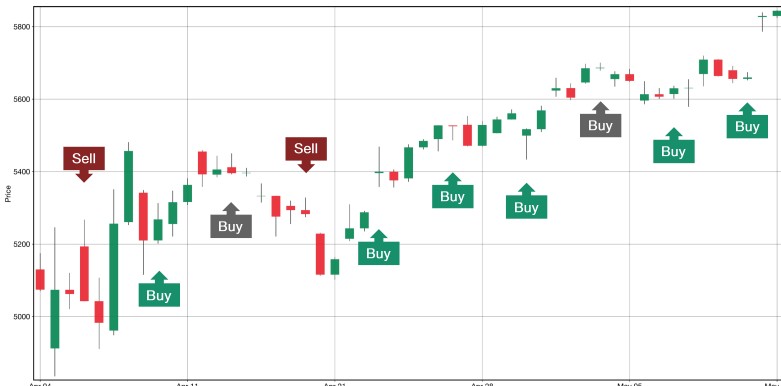

Figure 6: **Case study of high-frequency prediction on SPX (2025).** Correct forecasts (8/10) are marked with green Buy or red Sell badges, while mispredictions are shown in grey (2/10).

Furthermore, Figure 5 shows a comparative performance trend with same metrics for each asset in the 1-hour time frame. Our method (QuantAgent) consistently outperforms all baselines across most metrics and markets, especially in SPX, QQQ, and BTC where our method shows the most pronounced performance gap. The red line (QuantAgent) dominates across most plots, indicating both higher directional accuracy and better risk-aware returns, though it shows less satisfactory $\mathcal{R}_{\min}$ in some assets. Baseline, Linear Regression, and XGBoost exhibit weaker and less stable patterns, often lagging across most metrics. This visualization highlights the robustness and generalization capability of our approach under both profit-seeking and risk-constrained conditions in 1-hour time horizon.

## 5.2 CASE STUDY ON CONTINUOUS SHORT-TERM PREDICTION

To evaluate short-horizon prediction consistency, the LLM's directional accuracy was further tested on a randomly selected 100-bar SPX segment using 10 overlapping windows, each offset by 5 bars (Qin et al., 2017). Predictions were verified against actual price trends, achieving an overall accuracy of 80%, as shown in Figure 6. The system correctly issued sell signals at indices 0 and 3 during resistance stalls and negative momentum shifts, and buy signals at indices 1, 4, 5, 6, 8, and 9 by detecting early momentum flips, support bounces, and recognizable recovery patterns. Errors at indices 2 and 7 were due to overreliance on incomplete patterns and premature bullish calls, highlighting the model's tendency to prioritize emerging signals. Adjusting these weightings could enhance reliability.

## 5.3 CASE STUDIES OF AGENT REASONING

We present a representative Descending Triangle case study to illustrate PatternAgent's reasoning. Figure 4 illustrates how PatternAgent reasons over bar geometry. In this case, it recognizes a Descending Triangle, producing structured outputs that capture lower highs over flat support, a bearish breakdown bias, and triangular convergence. This example highlights the agent's ability to decompose raw price action into interpretable features. See Appendix E for additional case studies.

## 6 CONCLUSION

QuantAgent illustrates how decomposing trading into specialized LLM agents grounded in price data enables accurate, transparent, and risk-aware decisions for high-frequency trading. The multi-agent structure not only enhances interpretability but also promotes robustness through cross-agent validation and specialization. Our results across diverse markets underscore the viability of this approach, suggesting that structured agent collaboration grounded purely in price data can serve as a scalable foundation for future real-time, data-efficient financial systems operating without external sentiment or supervision.

## 7 REPRODUCIBILITY STATEMENT

We have taken multiple steps to ensure the reproducibility of our results. All datasets used in this work are available in an anonymous repository with link provided in abstract, and the benchmark construction is detailed in Appendix D, including asset coverage, sampling protocol, and data source. The implementation details of our system—including technical indicator computation, pattern extraction tools, trend detection methods, risk management, and the decision-making policies—are described in Section 3, with full agent prompt templates and sample tool outputs provided in Appendix B. In addition, we provide the source code and a web-based local demo interface in the same anonymous repository. Sample output of the web interface is included in Appendix C. Together, these resources allow our results to be independently verified and extended across new assets or timeframes.

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

## A WHY TECHNICAL ANALYSIS ALONE CAN SUFFICE FOR TRADING

QuantAgent is a multi-modal, multi-agent high-frequency trading LLM framework that provides market prediction based solely on price data, disregarding other information such as news, social media, etc. This strategy is referred to as technical analysis (Pring, 1991). Technical analysis is based on the premise that price alone is enough for capturing market movement and predicting future trends, and has been extensively studied by previous research (Murphy, 1999). In this section, we present in detail why technical analysis alone can suffice for trading.

The first principle of technical analysis is that all relevant information, whether economic, political, psychological, or otherwise, is already reflected in market prices (Fama, 1970). In other words, prices adjust quickly to new developments because people act on the information they receive by buying or selling (Kahneman & Tversky, 1979). These actions are recorded in price changes (Lo et al., 2000). Therefore, by observing how prices move, it is possible to indirectly understand how the market as a whole is reacting to both public and private information, without needing to process that information explicitly (Edwards et al., 2018). Our system therefore also follows this principle and has each of its agents perform analysis solely based on price data.

Technical analysis assumes that price movements are not entirely random (Lo et al., 2000). Instead, they tend to follow patterns over time. When prices begin to rise, they often continue to rise for some period, and similarly, downward trends can persist before reversing. These trends often reflect collective human behavior, such as fear during declines or optimism during rallies. By identifying such trends early, technical traders aim to make decisions that align with the general direction of the market (Jegadeesh & Titman, 1993). QuantAgent operates such that it captures market patterns and leverages this price movement assumption.

Notably, many existing technical analysis methods are based on the observation that certain price patterns tend to appear repeatedly. This repetition is attributed to stable behavioral tendencies in market participants. For example, traders often react similarly to price increases or decreases, leading to recurring patterns such as peaks, dips, and reversals. Recognizing and responding to these familiar structures allows technical systems to make predictions without needing to understand the specific causes of each movement (Edwards et al., 2018). Such observable repetition is a natural fit for an agentic framework, as LLM agents have shown strong capability in reasoning over patterns and trends, achieving human-like capabilities (Bommasani et al., 2021).

Occasionally, price changes occur well before any official information is made available to the public (Chakravarty et al., 1998). For example, a stock's price might begin rising days or even weeks before a company announces strong earnings. This can happen because certain investors—such as employees, suppliers, or professional analysts—may already have insights into the company's performance, such as increased sales activity or unusually high production volumes. Similarly, prices may fall before news of a scandal becomes public. If there are rumors of legal investigations or unusual management behavior, informed traders might start selling early, causing the price to decline in advance. In both cases, the price moves ahead of the news because the market collectively reacts to early signals, expectations, or private information (Chakravarty et al., 1998). Technical analysis captures these movements directly through price behavior, without requiring access to the underlying cause (Brock et al., 1992). This allows trading systems to respond to changes as soon as they appear in the market, rather than waiting for delayed or incomplete public disclosures (Lo et al., 2000).

In summary, our agent works under the principles of technical analysis, which offers a practical and self-contained approach to understanding market behavior. By assuming that all available information is already incorporated into price data, and that human reactions to price movements tend to follow consistent patterns, it becomes possible for our agent to forecast future price directions without relying on external inputs (Murphy, 1999).

## B PROMPT TEMPLATE

### B.1 INDICATOR AGENT

To operationalize the role of IndicatorAgent, we design a task-specific prompt that guides the agent to extract and interpret technical indicators under strict latency constraints.

---

### Prompt

You are a high-frequency trading (HFT) analyst assistant working under strict latency constraints.

You must analyze technical indicators to support fast-paced trading execution.

You have access to tools:
- compute_rsi

- compute_macd

- compute_roc

- compute_stoch

- compute_willr

Use them by providing appropriate arguments like 'kline_data' and the respective periods.

The OHLC data provided is from a {{time_frame}} interval, reflecting recent market behavior.

You must interpret this data quickly and accurately.

Here is the OHLC data: {{kline_data}}.

Call necessary tools, and analyze the results.

---

**Prompt for IndicatorAgent in our multi-agent LLM framework.** The agent receives recent OHLC data as input and interprets it through tool-assisted analysis. Variables such as kline_data and time_frame are dynamically instantiated, enabling the agent to extract meaningful price movements and adapt its outputs across diverse market conditions.

## B.2 PATTERN AGENT

To instantiate the PatternAgent, we construct a prompt that directs the agent to identify geometric formations (e.g., peaks, troughs, consolidations) from OHLC sequences, leveraging the LLM's multi-modal reasoning capacity for candlestick and chart-pattern analysis.

---
**Prompt**

You are a trading-pattern recognition assistant tasked with identifying classical high-frequency trading patterns.

You have access to tool: generate_kline_image.

Use it by providing appropriate arguments like 'kline_data'.

Once the chart is generated, compare it to classical pattern descriptions and determine if any known pattern is present.

...

---

**Prompt for PatternAgent in our multi-agent LLM framework.** The agent receives OHLC data as input, transforms it into a visual representation, and analyzes the sequence from a pattern-recognition perspective.

---
**Prompt**

This is a `{{time_frame}}` candlestick chart generated from recent OHLC market data.
Please refer to the following classic candlestick patterns:

1. Inverse Head and Shoulders: Three lows with the middle one being the lowest; symmetrical structure, typically precedes an upward trend.

2. Double Bottom: Two similar lows with a rebound in between, forming a "W".

3. Rounded Bottom: Gradual decline followed by a gradual rise ("U"-shape).

4. Hidden Base: Horizontal consolidation followed by a sudden up-break.

5. Falling Wedge: Range narrows downward, often resolves upward.

6. Rising Wedge: Range narrows upward, often resolves downward.

7. Ascending Triangle: Rising support, flat resistance; breakout usually up.

8. Descending Triangle: Falling resistance, flat support; breakout usually down.

9. Bullish Flag: Sharp rise then brief downward channel before continuation.

10. Bearish Flag: Sharp drop then brief upward channel before continuation.

11. Rectangle: Sideways range between horizontal support/resistance.

12. Island Reversal: Two gaps in opposite directions forming an "island".

13. V-shaped Reversal: Sharp decline followed by sharp recovery (or vice versa).

14. Rounded Top / Bottom: Gradual peaking or bottoming, arc-shaped.

15. Expanding Triangle: Highs and lows spread wider, volatile swings.

16. Symmetrical Triangle: Highs and lows converge; breakout after apex.

Determine whether the chart matches any of these patterns.

Name the detected pattern(s) and justify your choice based on structure, trend, and symmetry.

---

**Graph-analysis prompt for PatternAgent in our multi-agent LLM framework.** The agent is provided with a tool-generated chart and a textual library of canonical structural patterns (e.g., "U" shapes, "W" shapes, triangles). It automatically evaluates whether the chart matches any of these patterns and explains its reasoning along three dimensions: structure, direction, and symmetry.

## B.3 TREND AGENT

For the TrendAgent, we provide a prompt that emphasizes detection of directional momentum across multiple horizons, enabling the agent to reason about medium- to long-term trends while remaining responsive to short-horizon signals.

---

### Prompt

You are a K-line trend-pattern recognition assistant operating in a high-frequency trading context.

You must first call the tool 'generate_trend_image' using the provided 'kline_data'.

Once the chart is generated, analyze the image for support/resistance trendlines and known candlestick patterns.

Only then should you proceed to make a prediction about the short-term trend (upward, downward, or sideways).

Do not make any predictions before generating and analyzing the image.

---

**Prompt for TrendAgent in our multi-agent LLM framework.** The agent converts time-series OHLC data into a tool-generated chart and performs trend analysis on the visualization to identify directional momentum and potential breakouts.

---

### Prompt

You are a K-line trend-pattern recognition assistant in a high-frequency trading context. The following `{{time_frame}}` candlestick chart includes two automated trendlines: blue line is support, red line is resistance, both derived from recent closing prices.

Analyze how price..., are candles bouncing off, breaking through, or compressing between them?

Based on trendline slope..., predict the likely short-term trend: upward, downward, or sideways.

Support your prediction with respect to prediction, reasoning, signals.

---

**Graph-based prompt for TrendAgent in our multi-agent LLM framework.** The agent is provided with a tool-generated chart containing two reference lines: a blue support line and a red resistance line. It analyzes how price action interacts with these lines and produces a short-term trend prediction (upward, downward, or sideways), accompanied by structured outputs covering prediction, reasoning, and signals.

## B.4 DECISION AGENT

To implement the DecisionAgent, we design a prompt that compels the DecisionAgent to integrate signals from all specialized agents into coherent trading actions, balancing profitability, risk, and interpretability in high-frequency market settings.

---

### Prompt

You are a high-frequency quantitative trading (HFT) analyst reviewing the current {{time_frame}} K-line chart for {{stock_name}}.

Issue an immediate execution order: LONG or SHORT. (HOLD is prohibited.)

Forecast horizon. Predict price direction for the next 3 candlesticks (e.g. 15-min chart → next 45 minutes; 4-hour chart → next 12 hours).

Here is a LaTeX-style refactored version of your original decision guideline, following the concise structure and tone you provided:

—

Base your decision on three reports:

1. Technical Indicator Report — Evaluate momentum (MACD, ROC) and oscillators (RSI, Stochastic, Williams %R). Prioritize strong signals (e.g., MACD cross, RSI divergence, extreme levels). Down-weight mixed or neutral indicators unless aligned across types.

2. Pattern Report — Act only on clearly formed bullish/bearish patterns with breakout or breakdown confirmation (e.g., strong wick, volume spike, engulfing). Ignore early-stage or consolidating setups without support from other reports.

3. Trend Report — Analyze price interaction with trendlines: Up-sloping support = buying interest; down-sloping resistance = selling pressure. For compression zones, act only with clear candle or indicator confluence. Do not assume breakout direction from geometry alone.

Decision Strategy:

- Act only on confirmed, aligned signals across all three reports.

- Favour strong momentum and decisive price action (e.g., MACD crossover, rejection wick, breakout candle).

- If reports conflict, choose the side with stronger, more recent confirmation.

- In consolidation or unclear setups, defer to dominant trendline slope (e.g., short in descending channel).

- Do not speculate — choose the more defensible side.

- Suggest a risk-reward ratio between 1.2 and 1.8, adjusting for volatility and trend strength.

—

Let me know if you want this formatted directly in LaTeX code or exported to PDF.

Output JSON:

```
{
  "forecast_horizon" : "Predicting next N candlesticks (specify)",
  "decision"         : "<LONG or SHORT>",
  "justification"    : "<Concise confirmed reasoning>",
  "risk_reward_ratio": "<1.2 - 1.8>"
}
```

**Prompt for DecisionAgent in our multi-agent LLM framework.** The agent integrates three upstream perspectives, indicator signals, structural patterns, and trend interactions, and outputs a short-term directional decision (LONG or SHORT). The prompt instructs the agent to prioritize consistent evidence, avoid speculative outputs, and provide structured justification, including an estimated risk–reward ratio.

## B.5 Pattern Tool Sample Output

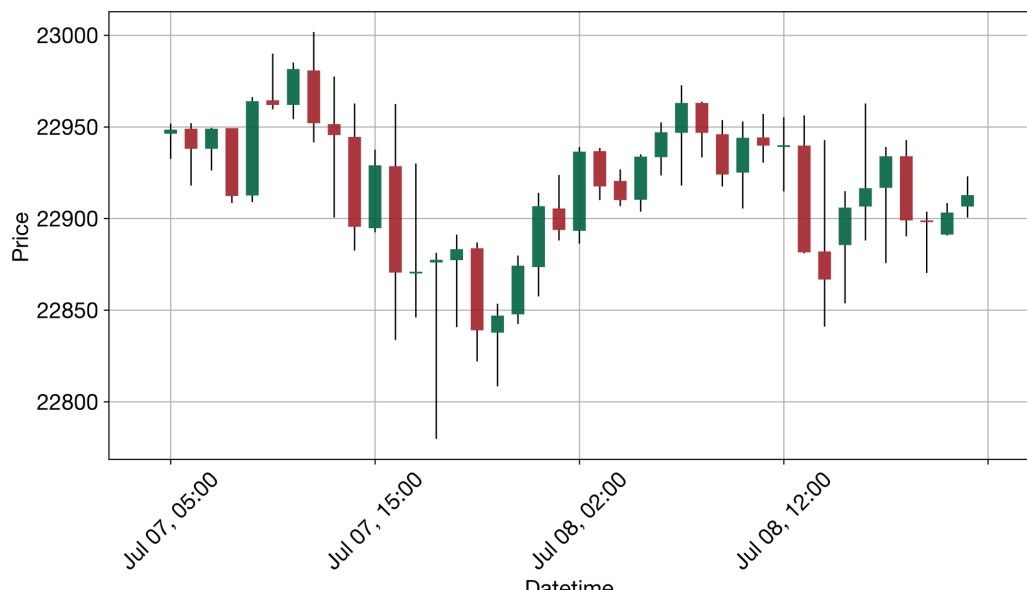

Figure 7: **Tool-generated chart for PatternAgent on NQ (2025).** Raw intraday candlesticks from the July 7–8 window are shown prior to overlaying reference lines. The sequence of lower highs and higher lows indicates a contracting trading range, suggesting latent pressure that often precedes a breakout once a boundary is breached.

## B.6    TREND TOOL SAMPLE OUTPUT

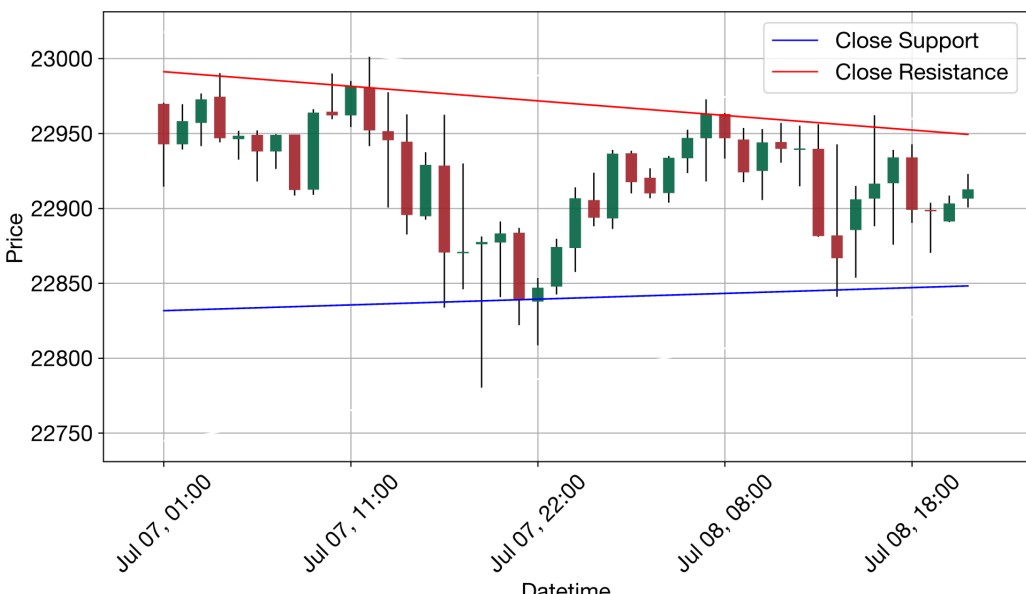

Figure 8: **Tool-generated chart for TrendAgent on NQ (2025).** Intraday candlesticks compress between an upward-sloping support line (blue) and a downward-sloping resistance line (red), forming a symmetrical-triangle wedge. The converging boundaries indicate a consolidation phase where buying pressure gradually builds while sellers cap rallies, often preceding a decisive breakout once a boundary is breached.

## C  WEB DEMO

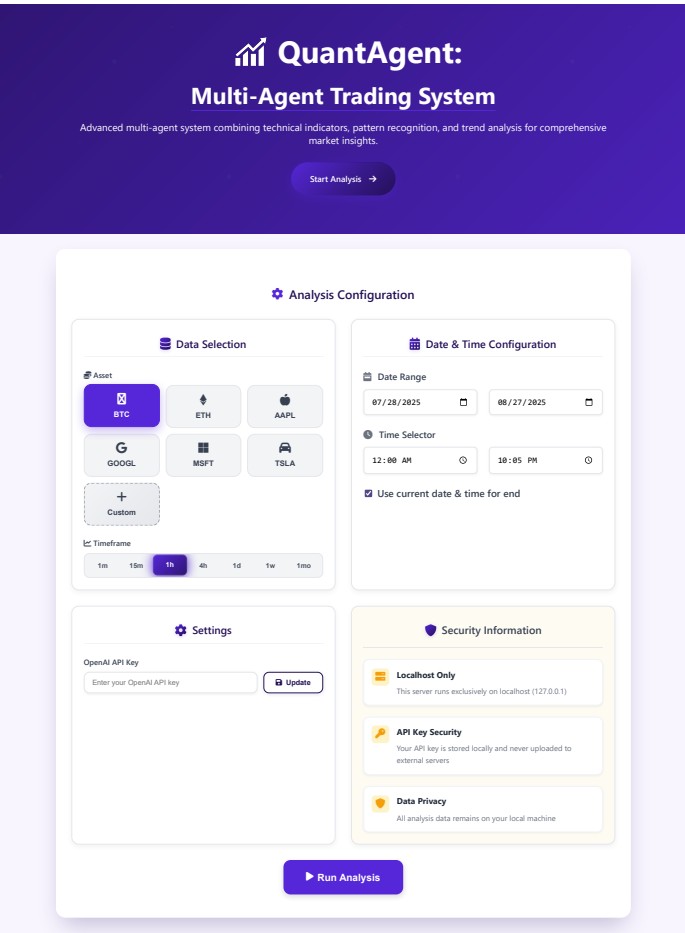

Figure 9: **User interface of QuantAgent.** The configuration panel enables selection of trading asset (e.g., AAPL), timeframe, and analysis window. It supports live data input, fine-grained control over historical candlestick ranges, and secure local execution. By default, the system optimizes context using the most recent 40–50 price bars to balance relevance and computational efficiency.

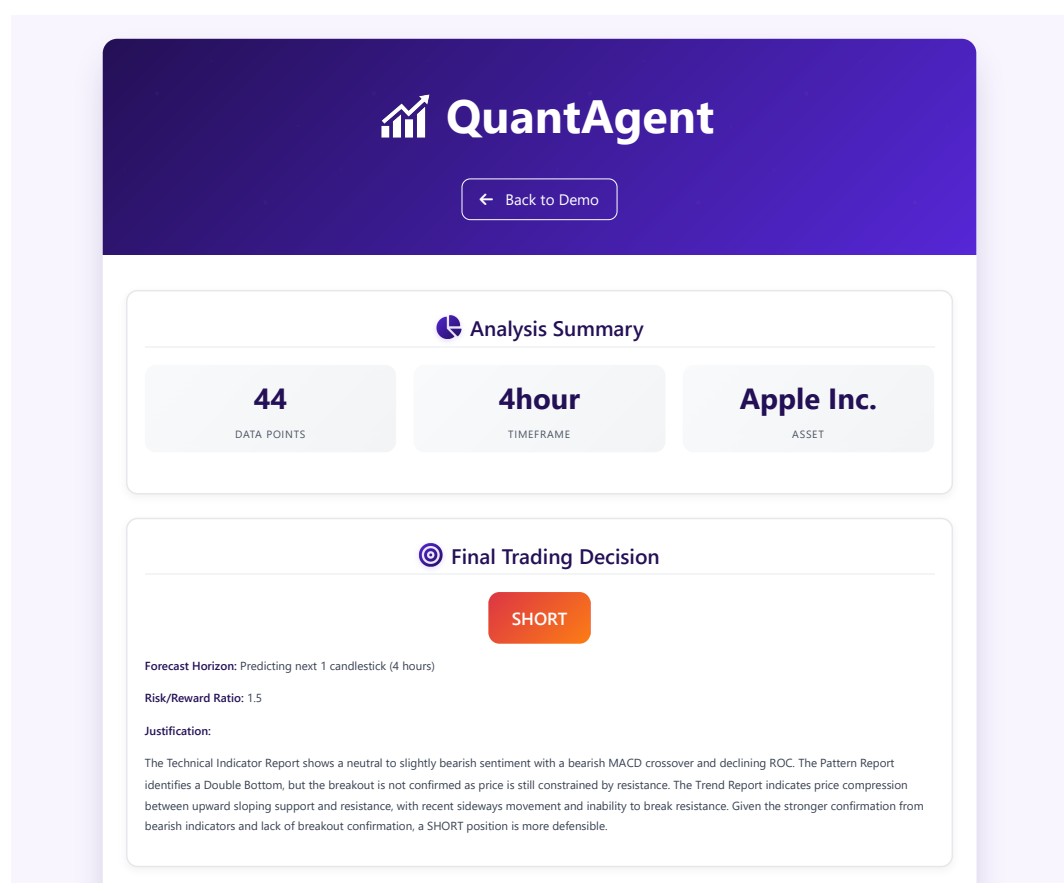

Figure 10: **Trading decision interface of QuantAgent.** The system produces a final directional decision along with the forecast horizon, risk–reward ratio, and a textual justification grounded in pattern recognition (e.g., Rounded Bottom reversal).

## 📊 Indicator Agent

Here's the analysis based on the technical indicators computed:

1. **Relative Strength Index (RSI):**

   ○ The RSI values are fluctuating between 46.09 and 71.54. The most recent RSI is 57.11, indicating a neutral to slightly bullish sentiment. RSI values above 70 typically indicate overbought conditions, while values below 30 indicate oversold conditions.

2. **Moving Average Convergence Divergence (MACD):**

   ○ The MACD line is currently at 2.72, with the signal line at 3.88. The MACD histogram is negative at -1.16, suggesting a bearish crossover as the MACD line is below the signal line. This indicates potential downward momentum.

3. **Rate of Change (ROC):**

   ○ The ROC values are showing a recent decline, with the latest value at -0.51. This suggests a slowing momentum in price increases, indicating potential bearish sentiment.

4. **Stochastic Oscillator:**

   ○ The %K line is at 45.92, and the %D line is at 41.43. Both lines are below 50, indicating a bearish sentiment. However, they are not in the oversold territory (below 20), suggesting there might still be room for further decline.

5. **Williams %R:**

   ○ The Williams %R is at -40.9, which is in the neutral zone. Values closer to -100 indicate oversold conditions, while values closer to 0 indicate overbought conditions. The current value suggests neither extreme.

### Summary:

The indicators collectively suggest a neutral to slightly bearish sentiment in the market. The RSI is neutral, the MACD indicates bearish momentum, the ROC shows slowing momentum, and the Stochastic Oscillator and Williams %R are not in extreme zones but lean towards bearishness. Traders might consider being cautious with long positions and look for potential short opportunities if the bearish signals strengthen.

Figure 11: **IndicatorAgent interface of QuantAgent.** A structured IndicatorAgent report is displayed, summarizing key technical indicators, MACD, RoC, Stochastic Oscillator, and Williams %R, to support interpretability and validate the decision process.

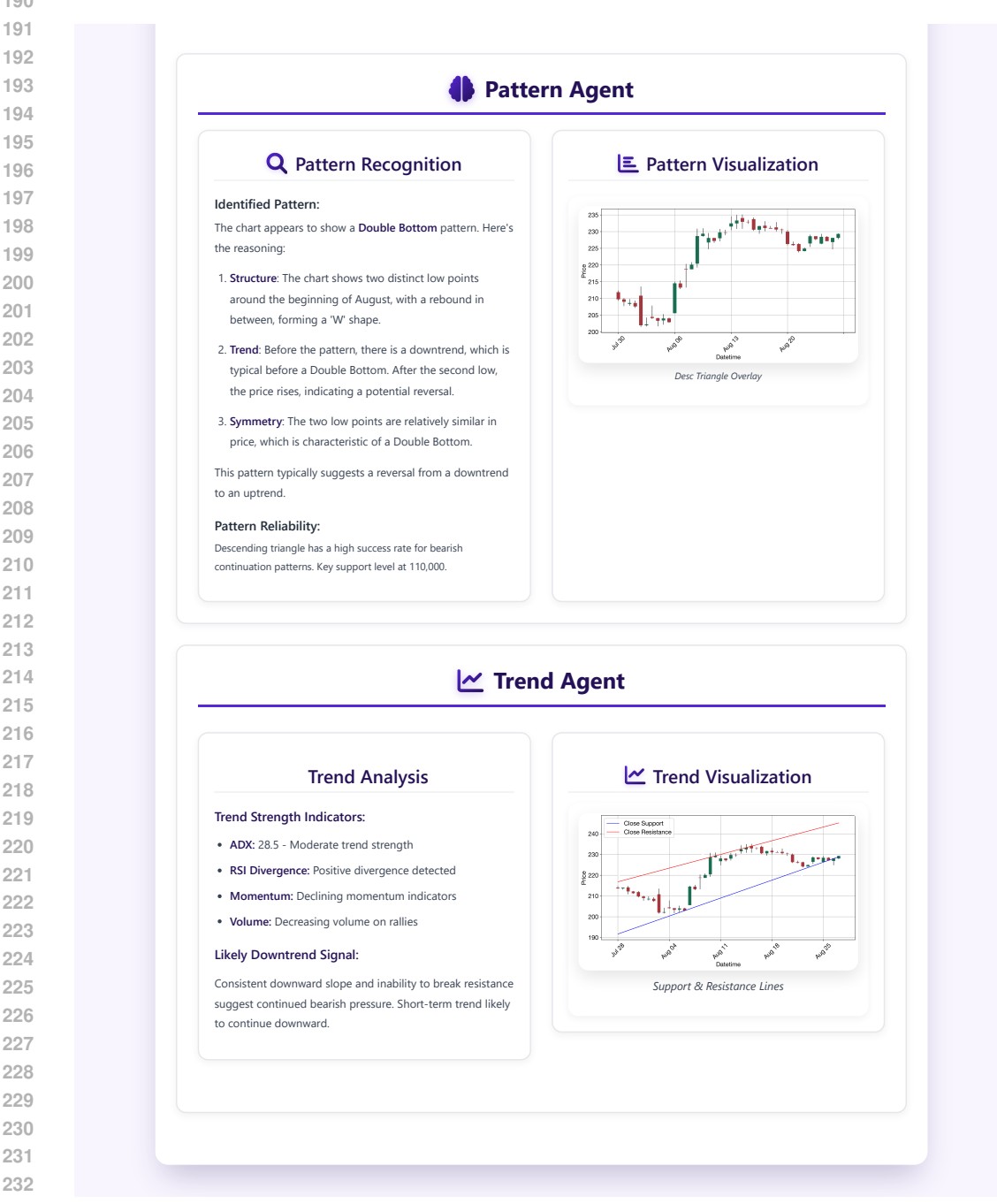

Figure 12: **Pattern and trend report generated by QuantAgent.** The top panel presents a detected Double Bottom pattern, supported by structural symmetry, a preceding downtrend, and subsequent rebound. The accompanying chart overlay highlights the pattern geometry. The bottom panel provides trend diagnostics, including ADX, RSI divergence, and volume analysis, alongside a visualization of support and resistance levels. Together, the pattern and trend modules offer complementary perspectives on potential trend reversal and market recovery.

## D  BENCHMARK DETAIL

### D.1  BENCHMARK CONSTRUCTION

To evaluate the proposed QuantAgent framework, we design a benchmark composed of diverse, well-known financial assets. This benchmark allows us to systematically test the system's ability to generalize across asset classes and trading environments. The benchmark also facilitates controlled comparisons across different decision-making models and enables reproducibility.

### D.2  DATA COLLECTION AND ASSET SELECTION

All historical price data used in the benchmark are obtained via the publicly available APIs, specifically YFinance and TradingView's historical market data services. We use 1-hour and 4-hour OHLC (Open, High, Low, Close) candlestick data for all assets to maintain consistency in time resolution. The benchmark covers a diverse mix of asset types, including cryptocurrency (BTC/USD), crude oil (CL), equity index futures (ES and NQ), and exchange-traded or spot indices such as QQQ, SPX, DJI, and VIX. Each asset is widely traded and highly liquid, helping avoid noise from low trading activity and making sure the price movements reflect real market behavior. Besides, the selected assets include both relatively stable, trend-following instruments such as SPX and ES, which often exhibit smoother directional movement, and more volatile assets such as BTC/USD, which are known for rapid swings and high short-term variability.

For each asset, we collect 5,000 historical 1-hour and 4-hour bars. To ensure fairness and consistency across assets, we apply the same fixed bar count to all instruments—including those with limited trading hours, such as QQQ. As a result, assets with lower intraday trading frequency span a longer historical period (up to ten years), reflecting the reduced density of available candlestick intervals. From this data, we randomly sample 100 evaluation segments per asset. Each segment consists of 100 consecutive candlesticks, with the final 3 candlesticks removed during inference to ensure the system does not observe the true market outcome during prediction. The final three candlesticks are reserved for validating the correctness of the LLM's prediction.

### D.3  BENCHMARK ASSET PROPERTIES

| Asset | Market Type | Start Date | End Date | Total Bars |
|---|---|---|---|---|
| BTC/USD | Cryptocurrency | 2023-04-01 | 2025-06-23 | 5000 |
| CL (Crude Oil) | Commodity Futures | 2022-04-25 | 2025-06-19 | 5000 |
| DJI | Equity Index (Spot) | 2015-08-26 | 2025-05-16 | 5000 |
| ES (S&P 500) | Equity Index Futures | 2022-04-19 | 2025-06-19 | 5000 |
| NQ (Nasdaq) | Equity Index Futures | 2022-04-19 | 2025-06-19 | 5000 |
| QQQ | Equity ETF | 2015-08-24 | 2025-05-16 | 5000 |
| SPX | Equity Index (Spot) | 2015-08-25 | 2025-05-16 | 5000 |
| VIX | Volatility Index (CBOE) | 2020-10-20 | 2025-08-27 | 5000 |

Table 2: Overview of 4-hour benchmark asset properties. Each asset is characterized by its name, market type, the start and end dates of the data collection window, and the total number of bars sampled.

| Asset | Market Type | Start Date | End Date | Total Bars |
|---|---|---|---|---|
| BTC/USD | Cryptocurrency | 2025-02-21 | 2025-09-13 | 5000 |
| CL (Crude Oil) | Commodity Futures | 2024-11-12 | 2025-09-10 | 5000 |
| DJI | Equity Index (Spot) | 2022-11-14 | 2025-09-02 | 5000 |
| ES (S&P 500) | Equity Index Futures | 2024-11-11 | 2025-09-10 | 5000 |
| NQ (Nasdaq) | Equity Index Futures | 2024-11-11 | 2025-09-10 | 5000 |
| QQQ | Equity ETF | 2022-11-14 | 2025-09-02 | 5000 |
| SPX | Equity Index (Spot) | 2022-11-14 | 2025-09-02 | 5000 |
| DAX | Equity Index (Futures) | 2024-10-21 | 2025-09-22 | 5000 |

Table 3: Overview of 1-hour benchmark asset properties. Each asset is characterized by its name, market type, the start and end dates of the data collection window, and the total number of bars sampled.

## D.4 BENCHMARK ASSETS DETAIL

We evaluate our models on a diverse set of benchmark assets drawn from major areas of the global financial markets.

BTC/USD (Bitcoin) shows how much one Bitcoin is worth in U.S. dollars. It represents the broader cryptocurrency market and operates continuously with high trading volume.

CL (Crude Oil) tracks the price of West Texas Intermediate crude oil, a key benchmark for U.S. energy prices and a global indicator influenced by supply, demand, and geopolitical factors.

ES (E-mini S&P 500) is a futures contract tied to the S&P 500 Index, which includes 500 large publicly traded U.S. companies. It gives a broad picture of the U.S. stock market's performance.

NQ (E-mini Nasdaq-100) follows the Nasdaq-100 Index, which focuses on large non-financial companies listed on the Nasdaq exchange, especially in the technology and innovation sectors.

QQQ is an exchange-traded fund (ETF) that aims to match the performance of the Nasdaq-100 Index. It offers a simple way for investors to gain exposure to major U.S. tech and growth stocks.

SPX (S&P 500 Index) directly tracks the S&P 500 Index and is widely used as a benchmark for measuring the overall performance of U.S. equities.

DJI (Dow Jones Industrial Average) includes 30 large and well-known U.S. companies across different industries. It is commonly used as an indicator of the broader U.S. economy.

VIX (Volatility Index) reflects the market's expectation of near-term volatility, often referred to as the "fear gauge" and widely used by investors to assess risk sentiment during periods of market uncertainty.

DAX (DAX Volatility Index) represents the market's expectation of short-term volatility in the German equity market. It is widely monitored by investors to assess risk sentiment and uncertainty surrounding the DAX 40 Index and broader eurozone conditions.

## D.5 CONCLUSION

This benchmark offers a consistent and comprehensive setting for evaluating trading systems across a range of asset classes. By standardizing the data resolution and segment format, it ensures fair and reproducible comparisons while still capturing the variety found in real-world markets. The inclusion of both stable, trend-following assets and more volatile instruments enables meaningful stress testing of model performance within multi-agent high-frequency trading frameworks like QuantAgent.

# E CASE STUDIES

When presented with the unannotated K-line window in Figure 4, the Pattern Agent first extracts swing pivots from recent bars and ranks successive local highs. The resulting pivot sequence forms a monotonic decline; a least-squares fit through those highs yields a negatively sloped resistance trajectory. In parallel, repeated lows cluster within a narrow tolerance band near the 78 price mark, producing an effectively horizontal support shelf. The vertical distance between the declining highs and flat support narrows over time, flagging range compression characteristic of a Descending Triangle. From these primitives the agent composes its three summaries: the `Structure Summary` reports "lower highs" over "relatively flat support"; the `Trend Summary` maps the recognized class to its empirical bearish bias, heightened breakdown probability once support is retested multiple times; and the `Symmetry Summary` abstracts the converging lines into a triangular shape descriptor used downstream for trigger/invalid level setting. The post-figure callouts (Lower Highs, dashed triangle edges, EMA overlays) are illustrative aids added for the reader; the pattern classification itself arises strictly from the bar-geometry analysis described above.

When the Trend Agent ingests the recent close-anchored K-line window in Figure 13, it fits a two-sided price channel to the rolling closes: a positively sloped upper boundary (resistance) and a relatively flat lower boundary (support). Repeated touches and near-touch rejections along the upper fit, combined with a long gap since the last interaction with support, signal that price action is track-

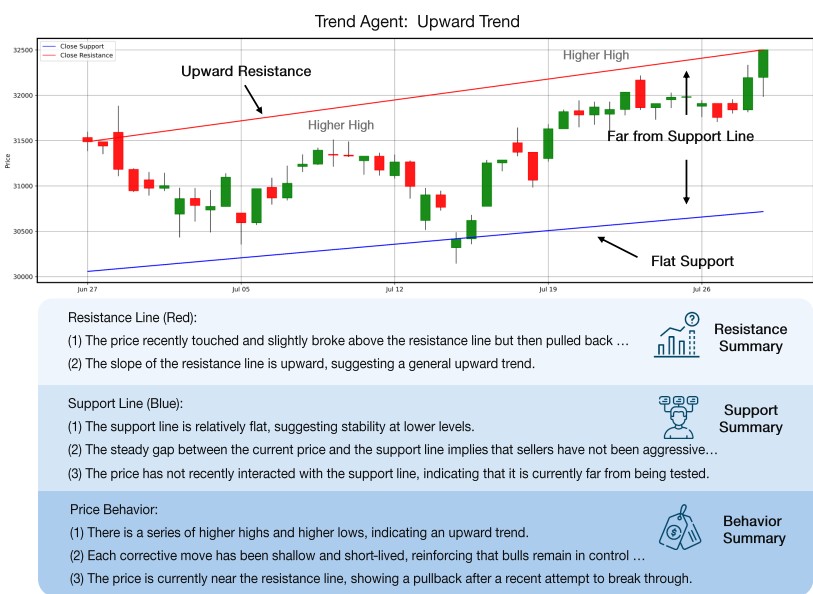

Figure 13: **Case sample of the `TrendAgent` on DJI (2022).** The agent fits an upward-sloping resistance line and flat support to recent closes, confirming successive higher highs and higher lows. It then generates three structured summaries: Resistance Line (upward slope), Support Line (untested flat base), and Price Behavior (higher highs/lows with pullback). The green "Bullish Signal" badge denotes the agent's final bullish assessment.

ing the upper rail of an advancing channel rather than oscillating symmetrically about its midline. The agent also tallies successive swing highs and lows; the resulting sequence is net higher, reinforcing an `upward trend` classification. These geometric diagnostics are distilled into the three text panels shown beneath the chart: (1) the `Resistance Line` summary notes a recent probe above the red boundary and its upward slope; (2) the `Support Line` summary records that price is far from a relatively flat blue base, implying untested downside room; and (3) the `Price Behavior` summary highlights the cluster of higher highs/higher lows and a pullback after resistance contact. The green "Bullish Signal" callout in the figure reflects this composite assessment: trend up, price extended near resistance, watch for either a breakout continuation or a tactical pullback entry toward support.

---

## IndicatorAgent Case Sample

Context: Latest OHLC snapshot (past time_frame).

Relative Strength Index (RSI). The RSI values are mostly above 50, indicating a generally bullish trend. However, the RSI has been fluctuating, with recent values around 68.49, suggesting the market is approaching overbought conditions.

Moving Average Convergence Divergence (MACD). The MACD line is above the signal line, particularly in the recent data, indicating a bullish momentum. The MACD histogram is positive, which supports the bullish sentiment.

Rate of Change (ROC). The ROC values are positive, indicating upward momentum. The recent ROC value of 0.94 suggests continued bullish momentum.

Stochastic Oscillator. The %K and %D lines are both above 80, indicating overbought conditions. This suggests that while the market is bullish, it may be due for a correction or consolidation.

Williams %R. The Williams %R values are mostly above -20, indicating overbought conditions. This aligns with the Stochastic Oscillator, suggesting caution as the market may be overextended.

Conclusion. The overall sentiment is bullish, with several indicators suggesting strong upward momentum. However, the overbought conditions indicated by the Stochastic Oscillator and Williams %R suggest that traders should be cautious of potential pullbacks or consolidations. It may be prudent to monitor for any signs of reversal or weakening momentum before making significant trading decisions.

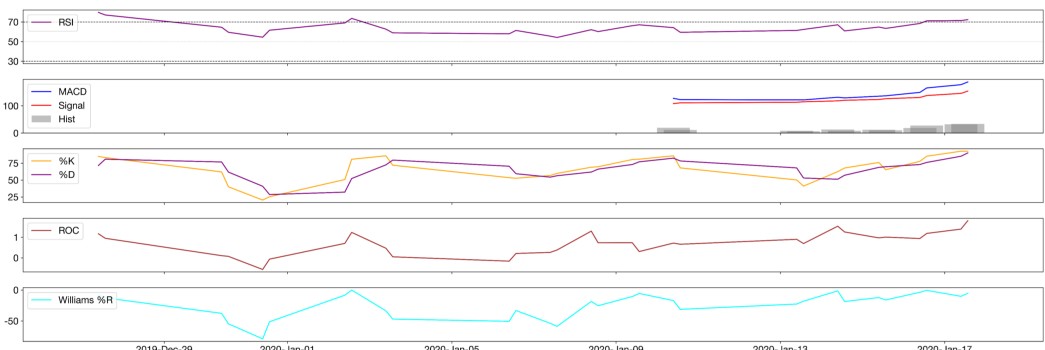

Figure 14: **Case sample of the IndicatorAgent on DJI (2019–2020).** From top to bottom, the panels show: (i) the four-hour RSI with neutral (50) and overbought thresholds, (ii) the MACD line and its signal line with histogram, (iii) the Stochastic %K/%D oscillator, (iv) the RoC rate-of-change, and (v) Williams %R. This multi-panel visualization presents the raw indicator series that define the momentum and oscillator primitives underlying the agent's "bullish but extended" assessment.

Given the latest OHLCV window, the Indicator Agent applies its momentum/oscillator transform suite (RSI, MACD, ROC, Stochastic, Williams %R) and aggregates the resulting state vector into a concise risk read. RSI has held above the neutral 50 line for most of the lookback and is presently in the high–60s ($\sim 68$), signaling sustained upside participation but proximity to the classic overbought band. MACD remains above its signal line with a positive histogram bulge, confirming that upside momentum is still in force. ROC readings are modestly positive ($\approx 1\%$), reinforcing a persistent upward rate of change rather than an exhausted spike. At the same time, fast oscillators cluster in warning territory: both Stochastic %K/%D and Williams %R sit in overbought zones ($> 80$ and $> -20$, respectively), indicating that the advance is stretched and vulnerable to a pause or mean reversion (Achelis, 2013). The agent therefore issues a composite summary of "bullish but extended": upside bias intact, yet tactical entries should respect exhaustion risk, tighten stops, scale position size, or await a reset toward support before adding exposure.

## F   INDICATOR EXPLANATION

Among the selected indicators, MACD is particularly representative due to its strengths in trend-following. MACD is designed to indicate momentum shifts by analyzing the divergence between two exponential moving averages (EMAs). It is calculated as:

$$\mathcal{E}_t = \alpha \cdot \mathcal{P}_t + (1 - \alpha) \cdot \mathcal{E}_{t-1} \qquad \mathcal{M}_t = \mathcal{E}_t^{(f)} - \mathcal{E}_t^{(s)} \qquad \mathcal{S}_t = \mathcal{E}_{\mathcal{M}_t}$$

The exponential moving average (EMA), denoted as $\mathcal{E}_t$, is a weighted average of past prices that assigns greater significance to more recent observations, thereby making it more responsive to recent price changes. Specifically, $\mathcal{P}_t$ represents the current price at time $t$, $\mathcal{E}_{t-1}$ is the EMA value from previous timestep, and $\alpha \in (0, 1)$ is the smoothing factor that determines the relative weight of the current price versus past EMA values. The factor $\alpha$ is typically computed as $\alpha = \frac{2}{N+1}$ where $N$ is the number of time periods (Achelis, 2013). Overall, EMA offers a smoothed representation of price trends, emphasizing recent movements while retaining historical context.

In our system, we define a fast EMA $\mathcal{E}_t^{(f)}$ with $N = 12$ and a slow EMA $\mathcal{E}_t^{(s)}$ with $N = 26$. The momentum signal $\mathcal{M}_t$ is then computed as the difference: $\mathcal{M}_t = \mathcal{E}_t^{(f)} - \mathcal{E}_t^{(s)}$ The signal line $\mathcal{S}_t$ is constructed as a 9-period EMA over the MACD sequence $\mathcal{E}_{\mathcal{M}_t}$: $\mathcal{S}_t = \mathcal{E}_{\mathcal{M}_t}$. A bullish signal occurs when $\mathcal{M}_t$ crosses above $\mathcal{S}_t$, indicating upward momentum, whereas a downward crossover suggests growing bearish pressure. This formulation makes $\mathcal{M}_t$ effective for capturing mid-term trend shifts (e.g., on 4-hour charts), while filtering out high-frequency price noise (Appel, 2005).

## G    THE USE OF LARGE LANGUAGE MODELS (LLMS)

Large Language Models (LLMs) were employed **only** as supportive instruments to enhance the readability and grammatical precision of our academic writing. In particular, GPT-4o was utilized to aid in refining portions of the manuscript, including the introduction and methodology. The authors maintain complete responsibility for the intellectual content, encompassing the formulation of research questions, the design of methods, and the verification of experimental findings.

