# OpenReview forum: "QuantAgent: Price-Driven Multi-Agent LLMs for High-Frequency Trading"
_ICLR.cc/2026/Conference — ICLR 2026 Conference Withdrawn Submission_

### Official Review · Reviewer_1PPc · 2025-10-15

**Soundness:** 1
**Presentation:** 3
**Contribution:** 2
**Rating:** 4
**Confidence:** 4

**Summary:**

This paper proposes QuantAgent, a price-driven multi-agent framework that integrates LLMs into high-frequency trading (HFT). The authors claim that multiple specialised agents (analyst, trader, and evaluator) coordinate to interpret market data, make trading decisions, and refine strategy iteratively. Experiments are conducted using cryptocurrency price data at one-hour (H1) and four-hour (H4) intervals, and results are compared against simple baselines such as random and moving-average strategies. The paper reports that QuantAgent achieves higher profit and win rate, suggesting that multi-agent reasoning can enhance short-term trading performance.

**Strengths:**

- The paper explores the potential of LLM-based multi-agent systems for quantitative trading.

- The experiments provide at least a basic quantitative evaluation, suggesting that LLMs can produce somewhat coherent trading signals.

- The paper is generally well-organised and easy to understand.

**Weaknesses:**

- The paper claims to study high-frequency trading, but the actual trading intervals (H1 and H4) are intraday or short-term horizons, not HFT. High-frequency trading typically involves sub-second decision cycles, co-location services, direct market access, and ultra-low-latency infrastructure, none of which are present here. Therefore, the “high-frequency” claim is misleading. The proposed setup resembles a conventional daily-trading or intraday backtesting framework, and if without the HFT label, some existing benchmark and baselines (such as FinMem, FinAgent) should be added to support the claim.

>Yu, Yangyang, et al. "Finmem: A performance-enhanced llm trading agent with layered memory and character design." IEEE Transactions on Big Data (2025).

>Zhang, Wentao, et al. "A multimodal foundation agent for financial trading: Tool-augmented, diversified, and generalist." Proceedings of the 30th acm sigkdd conference on knowledge discovery and data mining. 2024.

- Follow up the previous point, the framework still lacks proper baselines. Even ignoring the incorrect HFT framing, there are no comparisons against standard quantitative-trading strategies such as pairs trading, mean-reversion, or momentum-based methods, nor against reinforcement-learning-based trading agents. Without such baselines, it is impossible to gauge how the framework performs relative to established approaches.

- The evaluation design is weak. It does not include core backtesting metrics such as Sharpe ratio, Sortino ratio, or maximum drawdown, and ignores essential trading factors like commission fees, slippage, and liquidity limits. This omission makes the profitability results unreliable and economically meaningless.

- There is no report of latency or computational timing to justify the system’s claimed high-frequency capability. The entire “multi-agent” loop (analyst–trader–evaluator) could take seconds or minutes per decision, yet the paper provides no latency measurements or runtime analysis. Without this, the HFT claim collapses.

- There is no ablation or sensitivity analysis, and the multi-agent design lacks clear justification. The paper does not test how many agents are necessary, whether they provide complementary perspectives, or if a single-agent setup would yield comparable results. The agent roles appear arbitrary and unverified. This weakens the overall methodological credibility and suggests the “multi-agent” design may add unnecessary complexity without real gain.

**Questions:**

- What is the actual latency (in milliseconds or seconds) of a full agent loop from market data input to trade decision?

- Why are H1 and H4 intervals referred to as “high-frequency”? Please justify this terminology or revise the claim.

- Can the authors include standard baselines such as pairs trading, mean-reversion, or reinforcement-learning agents for fair comparison?

- Please incorporate standard financial metrics such as Sharpe ratio, Sortino ratio, and maximum drawdown to evaluate risk-adjusted returns.

- Could ablation experiments be added to test the necessity and contribution of each agent role?

---

### Official Review · Reviewer_GrVJ · 2025-11-01

**Soundness:** 1
**Presentation:** 3
**Contribution:** 2
**Rating:** 2
**Confidence:** 4

**Summary:**

This paper introduces QuantAgent, a multi-agent LLM framework designed for algorithmic trading. This framework contains several agents, including indicator agent, pattern agent, trend agent, risk agent and decision agent.

**Strengths:**

The agents desinged in this paper are useful indicators for the high-frequency trading task.
For the pattern and trend recognition, using VLM techniques is appropriate and efficient.

**Weaknesses:**

1. Fundamental Misnomer of "High-Frequency Trading". This is the paper's most significant flaw. The title and abstract claim the system is for HFT, which in finance implies microsecond-to-millisecond-level operations based on order book data. The experiments are conducted on 1-hour and 4-hour bars. This is not HFT; it is, at best, "intraday" or "short-term" algorithmic trading. This misrepresentation undermines the paper's framing, as the latencies of this system are completely unaddressed and would be orders of magnitude too slow for true HFT.

The paper completely ignores the computational cost and, more importantly, the end-to-end latency of this system. A pipeline that chains multiple LLM calls, including image generation and multimodal analysis, likely takes several seconds to run. This is a critical omission for a paper claiming relevance to high-speed trading.

2. Key Logic is Unexplained (Risk/Decision): The logic for the two most critical agents is poorly defined.
The RiskAgent "predicts" a risk-reward ratio from a fixed range of [1.2, 1.8]. How? This is a crucial parameter for profitability. Is it a random choice? Is the LLM prompted to "pick a number"? This is left entirely to the reader's imagination.
The DecisionAgent's prompt (Appendix B.4) provides high-level heuristics ("Act only on confirmed, aligned signals") but doesn't formalize how conflicts are resolved, which is the entire purpose of an aggregation agent.

3. The benchmark models. There are several classic model such as AS model and GLFT model in HFT. You may want to use these to compare the performance.

4.Unspecified Chart Generation Parameters: The patterns agent performance is fundamentally dependent on the chart it "sees." A critical parameter for this is the lookback window.

**Questions:**

What is the end-to-end latency for the full QuantAgent pipeline (from new bar data to final decision)? How does this latency impact the feasibility of applying this system to shorter timeframes (e.g., 1-minute or 5-minute bars)?

For the PatternAgent and TrendAgent, how is the lookback window selected for generating the charts? Is this a fixed hyperparameter (e.g., 50 bars), and if so, how was it optimized?

---

### Official Review · Reviewer_J7mQ · 2025-11-02

**Soundness:** 2
**Presentation:** 3
**Contribution:** 3
**Rating:** 4
**Confidence:** 3

**Summary:**

The paper proposes QuantAgent, a novel multi-agent LLM framework designed for high-frequency trading (HFT) using structured price signals (OHLC data) instead of traditional textual inputs. The system consists of four specialized agents (Indicator, Pattern, Trend, and Risk) that collaboratively interpret technical signals and generate risk-aware, interpretable trading decisions.

**Strengths:**

- Rigorous system architecture design; meticulous agent design; robust experimental setup incorporating missing data analysis and baseline comparisons.
- Pioneered a rarely explored domain—applying large language models for real-time, low-latency decision-making in high-frequency trading environments.
- Each agent generates interpretable outputs (e.g., decision rationale, pattern structures), significantly enhancing transparency compared to black-box models.

**Weaknesses:**

- While the system architecture is well-engineered, many components (e.g., technical indicators, regression-based trend detection) are standard and repurposed.
- Absence of latency or inference time benchmarks. Critical in HFT, yet not evaluated or discussed.

**Questions:**

- Are there latency and throughput metrics for QuantAgent during live inference? HFT performance depends heavily on timing.

- How does QuantAgent handle conflicting signals among agents (e.g., TrendAgent signals uptrend, but PatternAgent detects reversal)? Is there a confidence aggregation mechanism?

---

### Note · Authors · 2025-11-12

I have read and agree with the venue's withdrawal policy on behalf of myself and my co-authors.